# Chronic Alcohol Use Induces Molecular Genetic Changes in the Dorsomedial Thalamus of People with Alcohol-Related Disorders

**DOI:** 10.3390/brainsci11040435

**Published:** 2021-03-29

**Authors:** Andreas-Christian Hade, Mari-Anne Philips, Ene Reimann, Toomas Jagomäe, Kattri-Liis Eskla, Tanel Traks, Ele Prans, Sulev Kõks, Eero Vasar, Marika Väli

**Affiliations:** 1Department of Pathological Anatomy and Forensic Medicine, University of Tartu, 19 Ravila Street, 50411 Tartu, Estonia; andreas-christian.hade@ekei.ee (A.-C.H.); Marika.Vali@ut.ee (M.V.); 2Forensic Medical Examination Department, Estonian Forensic Science Institute, 30 Tervise Street, 13419 Tallinn, Estonia; 3Department of Physiology, Institute of Biomedicine and Translational Medicine, University of Tartu, 50411 Tartu, Estonia; toomas.jagomae@gmail.com (T.J.); kattriliis@gmail.com (K.-L.E.); eero.vasar@ut.ee (E.V.); 4Centre of Excellence in Genomics and Translational Medicine, University of Tartu, 50411 Tartu, Estonia; 5Estonian Genome Centre, Institute of Genomics, University of Tartu, 51010 Tartu, Estonia; Ene.Reimann@ut.ee; 6Department of Dermatology and Venerology, Institute of Clinical Medicine, University of Tartu, 51010 Tartu, Estonia; tanel.traks@ut.ee; 7Department of Anaesthesiology and Intensive Care, Tartu University Hospital, 50406 Tartu, Estonia; ele.prans@ut.ee; 8Perron Institute for Neurological and Translational Science, Perth, WA 6009, Australia; sulev.koks@perron.uwa.edu.au; 9Centre for Molecular Medicine and Innovative Therapeutics, Murdoch University, Perth, WA 6150, Australia

**Keywords:** alcoholism, alcohol use disorder, autopsy, mediodorsal thalamus, GRIN1, FTO

## Abstract

The Mediodorsal (MD) thalamus that represents a fundamental subcortical relay has been underrepresented in the studies focusing on the molecular changes in the brains of subjects with alcohol use disorder (AUD). In the current study, MD thalamic regions from AUD subjects and controls were analyzed with Affymetrix Clariom S human microarray. Long-term alcohol use induced a significant (FDR ≤ 0.05) upregulation of 2802 transcripts and downregulation of 1893 genes in the MD thalamus of AUD subjects. A significant upregulation of GRIN1 (glutamate receptor NMDA type 1) and FTO (alpha-ketoglutarate dependent dioxygenase) was confirmed in western blot analysis. Immunohistochemical staining revealed similar heterogenous distribution of GRIN1 in the thalamic nuclei of both AUD and control subjects. The most prevalent functional categories of upregulated genes were related to glutamatergic and GABAergic neurotransmission, cellular metabolism, and neurodevelopment. The prevalent gene cluster among down-regulated genes was immune system mediators. Forty-two differentially expressed genes, including FTO, ADH1B, DRD2, CADM2, TCF4, GCKR, DPP6, MAPT and CHRH1, have been shown to have strong associations (FDR *p* < 10^−8^) with AUD or/and alcohol use phenotypes in recent GWA studies. Despite a small number of subjects, we were able to detect robust molecular changes in the mediodorsal thalamus caused by alcohol emphasizing the importance of deeper brain structures such as diencephalon, in the development of AUD-related dysregulation of neurocircuitry.

## 1. Introduction

Alcohol is the most widely used addictive substance worldwide. Excessive alcohol use is associated with deleterious neurobehavioral consequences to the drinker and causes significant burden to the family members and for the society as a whole [1]. This burden results from the increased health risks associated with alcohol use disorder (AUD), as well as from the social harms caused by alcohol.

AUD is a complex disorder, with genetic and other biological factors involved in the pathways leading to its development. Alcohol use patterns, including harmful use and heavy episodic drinking, have a familial pattern in nature, with twin studies supporting this conclusion [2,3]. For the past few years, genome wide association studies (GWAS) of AUD and related traits such as alcohol consumption measurements, have been conducted by using the sample sizes exceeding hundreds of thousands of cases and controls [4,5,6,7,8,9]. These studies have been identifying a few dozen genes and related pathways associated with AUD enlightening the genetic predisposition increasing the risk of AUD.

Chronic alcohol use and the progression into dependence produces persistent neuronal alterations in the brain that are likely a consequence of altered gene and protein expression that underlie the cellular adaptations to chronic alcohol use [10,11]. Microarray technology and next-generation sequencing have changed the way, in which genes are studied, and proven to be valuable tools in the study of common, but genetically complex diseases, such as AUD, because they allow large numbers of transcribed elements to be examined simultaneously in an unbiased fashion. Identifying these expression differences may help us understand the biological causes of this disorder, the effects of its exposure on gene expression, as well as possibly identifying candidates, to examine a trait or phenotype, which could finally lead to better screening and treatment.

Most of the post-mortem human genome-wide gene expression profiling studies have been concentrated on studying alcohol-responsive transcriptional changes in the prefrontal cortex [12,13,14,15]. Other studies have also been studying hippocampus [16,17], nucleus accumbens [18] and amygdaloid nuclei [12]. In general, the studies on post-mortem human brain samples have concentrated on the limited areas of the human brain. Earlier studies suggest that alcohol may induce distinct alterations in the different areas across the human brain. Discrete effects of alcohol have been shown across the frontal cortex and distinct amygdaloid nuclei [12], in the nucleus accumbens compared to hippocampus [17,18], and in the frontal cortex compared to hippocampus [19].

The thalamus is integral to the circuits that underlie the reward and response inhibition processes mediating salience and control processes in goal-oriented behaviors (for a review, see Huang et al. [20]). Although the frontal cortex has been in the focus of the research seeking alcohol-induced molecular changes in the human brain, the functions of the forebrain are largely built on the properties of thalamic projection neurons making the thalamus the central communication hub of the forebrain [21]. Furthermore, alcohol-induced neurological damage is dependent on the sensitivity of thalamic nuclei [22,23]. The mediodorsal (MD) nuclei of the thalamus have extensive reciprocal innervation with the prefrontal cortex (PFC) with the MD providing the majority of excitatory innervation to the PFC. Abundant studies have focused on understanding the role of the subcortical MD network in the regulation of PFC-related cognitive processes, such as working memory, attention, and cognitive flexibility (for a review, see Ferguson & Gao [24]).

As the thalamus has been underrepresented in the studies of chronic alcohol induced molecular changes in the human brain, the purpose of the present study was to perform a hypothesis-free comparison of global gene expression profiles by conducting a microarray study of postmortem mediodorsal thalami of AUD and control subjects. We also compared our results with genes identified in recent genome-wide association studies (GWAS) for alcohol dependence or related phenotypes and validated selected transcriptional changes in the microarray by using western blot in the protein level. Our goal was to shed light on the chronic alcohol-induced molecular changes in the dorsomedial thalamus that is an important subcortical relay with well described impact in the formation and progress of alcohol induced neuronal damage.

## 2. Materials and Methods

### 2.1. Human Post-Mortem Brain Tissue Collection and Dissection

Brain tissues originating from 27 post-mortem examinations of Caucasian individuals were collected by the Estonian Forensic Science Institute. The bodies were stored in +4 °C and brought to the autopsy room just prior to the start of the examination. The chronic alcohol use of the deceased (AUD group, *n* = 14) was based on a prior diagnosis of alcohol related disorder together with clinical diagnosis of alcohol related end-organ damage supported by post-mortem histology, including alcohol-related liver disease, alcohol-related pancreatitis, and alcohol-related cardiomyopathy. The control group (*n* = 12) included subjects which had no prior diagnosis of alcohol related disorder code, end-organ damage, and signs of alcohol related damage collected from tissue samples taken during the autopsy. The subjects with liver cirrhosis were only included into the control group in case of confirmed clinical diagnosis of liver cirrhosis due to biliary cirrhosis or hepatitis C. Subjects with a known diagnosis of diabetes and/or prior history of ketoacidosis, including diabetic and/or alcohol-related, were excluded. Toxicological analysis was routinely performed and none of the selected subjects presented acetone levels indicating ketoacidosis. The average age was 47 ± 12.04 (mean ± SD) in the group of AUD, and 49.58 ± 10.05 in the group of control subjects. The material from 11 subjects (6 AUD and 5 controls) was chosen for genechip analysis; the material from different 12 subjects (6 AUD and 6 controls) was selected for western blot analysis (one control subject was chosen for both genechip and western analysis); the brain samples from 5 subjects (2 AUD and 3 controls) were selected for immunohistochemistry studies. Different study subjects were chosen for genechip analysis and western blot in order to confirm the same molecular changes in an independent study group and in order to and in order to provide evidence of the significance of microarray data in the protein level. The post-mortem interval (PMI) was calculated from the time of death to the time of removal of the brain from the skull and the average PMI in the sample collection was 31.92 ± 12.88 (mean ± SD) hours. The median warm time (time before cold storage at 4 °C) was 5.61 ± 3.13 h for AUD subjects and 6.75 ± 2.97 h for control subjects, therefore, most of the PMI represents cold time, which slows down postmortem tissue degradation [25]. More details about the study subjects are provided in Appendix A.

The dissections were performed by qualified pathologists under the full ethical approval from the Research Ethics Committee of the University of Tartu (Approval no 258/T-8 (18 April 2016) from human research ethics committee). For immunohistochemistry, the tissue blocks including medial dorsal and ventral lateral thalamic nucleus (MNI coordinates from −21.14 to −18.30 according to the Atlas of the Human Brain [26], Figure 1A) were immersion fixed in 4% paraformaldehyde (for further details see Section 2.6). For RNA and protein extraction, the mediodorsal thalamic nucleus was dissected from the tissue block, the samples were dissected into pieces of approximately 250–500 mg and frozen at −80 °C. Brain pH level was measured from the side ventricles using a Hanna HI-99163 hand-held pH meter with a glass bodied electrode. The electrode was inserted into the side ventricle after the brain was coronal cut behind the mammillary bodies. Since low brain pH is associated with prolonged agonal states [27], the samples with pH lower than 5.8 were excluded, the average pH in our sample collection was 6.258 ± 0.047 (mean ± SD).

### 2.2. RNA Extraction from Tissues and Whole Transcriptome Analysis

RNA was extracted by using the TRIzol™ Reagent (Thermo Fisher, Waltham, MA, USA) according to the manufacturer’s protocol. In short, up to 100 mg of brain tissue was homogenized by applying tissue grinders (Axygen™, Union City, CA, USA) and 1 mL of TRIzol™ Reagent, followed by a phase separation step where chloroform was used. RNA was precipitated with isopropyl alcohol. The quantity and quality of the RNA-extracts were monitored applying a NanoDrop 2000 Spectrophotometer (Thermo Scientific™, Waltham, MA, USA) and 2100 Bioanalyzer (Agilent Technologies) together with RNA 6000 Nano kit. The amount of RNA received from samples was between 21–71 μg, the A260/280 ratio was between 1.87–2.01, the A260/230 ratio was between 1.98–2.21, the average RIN value of RNA samples was 4.389 ± 0.456 (mean ± SD). Throughout the RNA extraction Eppendorf DNA/RNA LoBind tubes were applied (Sigma-Aldrich, St. Louis, MO, USA).

For each sample, 100 ng RNA input was taken for gene expression analysis. Gene expression profiling was conducted with Affymetrix GeneTitan Multi-Channel platform with Clariom S human microarray and GeneChip WT (whole transcriptome) Plus kit (Affymetrix Inc, Santa Clara, VA, USA) according to the manufacturer’s protocol. Clariom S microarray provides data for more than 20,000 well-annotated genes.

### 2.3. Differential Gene Expression Analysis

Differential gene expression analysis was performed with Transcriptome Analysis Console (TAC) software (Affymetrix Inc). TAC performs quality control and statistical analysis for differential expression and pathway analysis. R Bioconductor package limma is implemented in the TAC making the statistical framework R accessible for the users [28]. Limma provides a very complex and flexible analytical framework for genome-wide expression analysis [29]. We compared the RNA expression in thalamus tissues collected from AUD group versus controls. A False Discovery Rate (FDR) was applied for the correction of multiple testing [30] (Appendix A). TAC software was also applied for clustering analysis and preparing the Heatmap. The current analysis has methodological limitation which must be pointed out. Due to the limited access to post-mortem tissues, we were not able to perform a pilot study evaluating the potential amount of variation in the data, which is needed for the power calculations. Also, the minimal effect size, which would be clinically or scientifically meaningful, is hard to evaluate at the current knowledge stage. Thus, with a small sample available for us we expected to see an occurrence of medium or large effects in the thalamic area of AUD subjects versus controls.

### 2.4. Functional and Pathway Analysis

Following the differential gene expression analysis, we performed gene ontology analysis applying g:Profiler’s g:GOSt functional profiler tool, which is a web server for functional enrichment analysis and conversions of gene lists [31]. Statistical significance was based on g:SCS threshold suggested by g:Profiler, which is accounting for multiple comparisons in the analysis, with threshold less than 0.05. Differentially expressed genes (DEGs, Fold Change ≥ ±2, FDR < 0.01) used for heatmap clustering analysis (Figure 2) were used as input in the Gene Ontology analysis applying the g:GOSt tool. The GO molecular function, cellular component and biological process data sources were considered (Appendix A).

The full lists of DEGs (FDR ≤ 0.05), were then applied to the Gene Ontology GO enrichment analysis tool (The Gene Ontology Consortium [32]; http://geneontology.org/; accessed on 15 May 2020). The list of upregulated genes was submitted separately from the list of downregulated genes (Appendix A), annotations were based on GO “molecular function”.

Finally, we applied the WikiPathways analysis tool integrated into TAC for identification of dysregulated functional pathways (Appendix A). WikiPathways analysis is based on identification of differentially expressed functional pathways within a gene regulatory network using gene expression data analysis. WikiPathways contains information about biomolecules and their interactions that provides valuable context and fodder for data analysis and visualization [33]. Here, we used the full gene list (21,448 genes) as input and filtered out all the genes with significance values FDR ≤ 0.05.

### 2.5. The Comparison of Gene Expression Changes with Recent GWAS Studies

The differentially expressed transcripts were compared with the results of major recent GWAS studies related to the AUD and alcohol use phenotypes. The GWAS analysis and meta-analysis that included more than 100,000 study subjects published from 2019 to 2020 were included in the analysis [4,5,6,7,8,9,34,35]. The genes that gave significant genome wide (FDR *p* < 10^−8^) associations with alcoholism were compared with the list of genes significantly (FDR *p* < 0.05) altered in the dorsomedial thalamus of AUD subjects. In case the GWAS study data was included in the NHGRI-EBI Catalog of human genome-wide association studies (https://www.ebi.ac.uk/gwas/home; accessed on 15 December 2020), the GWAS Catalog tool was used to access GWAS data. The overlapping genes that were confirmed as significant in at least two GWAS studies are listed in Table 1.

### 2.6. Western Blot

The selected transcriptional changes in the microarray were chosen for further validation in protein levels by using western blot analysis. GRIN1 was chosen as the well characterized glutaminergic receptor [36], FTO is a gene of interest due to high significance associations with AUD and related phenotypes in the recent GWAS studies [4,5,8]. Cyt C was selected to represent upregulated transcripts related to cellular metabolism and GAPDH, which is a well-known house-keeping gene, was further tested as it showed significant upregulation in the microarray indicating its inappropriateness as a housekeeping gene in the studies of brain tissues of the subjects with alcohol-related disorders. Thalamus samples were sonicated in an ice-cold RIPA lysis buffer (89900, Thermo Scientific) supplemented with 1× protease inhibitor (78430, Thermo Scientific). The samples were incubated on ice for 30 min and centrifuged at 14,000× *g* for 10 min at +4 °C. Protein concentrations were determined using BCA protein assay kit (23225, Thermo Scientific). Equal amounts of protein (20 μg) were resolved on NuPAGE Bis–Tris gel and transferred to nitrocellulose membrane using the XCELL SureLock System (Invitrogen). The membrane was then incubated in a blocking buffer followed by incubation with primary antibodies (Appendix A) overnight at 4 °C. After washing, the membrane was probed with goat anti-rabbit (A11369, Invitrogen) or goat anti-mouse (A-21057, Invitrogen) fluorescent conjugated secondary antibodies for 1 h at room temperature, followed by visualization using a LI-COR Odyssey CLx system (LI-COR Biotechnologies, Lincoln, NE, USA). The images were converted to grayscale and the density of protein was quantified using Image Studio Lite v 3.1.4 (LI-COR Biotechnologies). B-actin was used as a loading control.

### 2.7. Immunohistochemistry and Fluorescence Intensity Measurements

Thalami were immersion fixed in 4% paraformaldehyde (PFA, Honeywell, Charlotte, NC, USA)/PBS (pH 7.4) for 48 h at 4 °C. Subsequently tissues were washed over several days with PBS/0.1% sodium azide (AppliChem, Darmstadt, Germany) and impregnated with 30% sucrose (Fisher Scientific, Waltham, MA, USA)/PBS/0.1% sodium azide solution until sunk. Thalami were frozen by immersing in 2-methylbutane (Honeywell) chilled on dry ice and kept at −80 °C until sectioning. The subsequent protocol for immunohistochemistry has been described previously [37], where the primary antibodies were diluted in 10% normal goat serum (NGS)/1% bovine serum albumin (BSA)/0.2% Triton X-100/PBS solution 72 h at 4 °C with gentle shaking. Primary antibodies and their dilutions were: GRIN1 (1:500, Abcam Cat# ab17345, RRID:AB_776808), NeuN (1:500, Millipore Cat# MAB377, RRID:AB_2298772). Polyclonal secondary antibodies and their dilutions in 1% BSA/0.2% Triton X-100/PBS were: goat anti-rabbit IgG H&L (Alexa Fluor 488) (1:500, Molecular Probes Cat# A-11094, RRID:AB_221544), goat anti-mouse IgG H&L (Alexa Fluor 555) (1:500, Abcam Cat# ab150114, RRID:AB_2687594)). Nuclei were counterstained with 5 μg/mL Bisbenzimide H 33258 (Hoechst 33258, Sigma Aldrich) in PBS. Images from the thalamic region were obtained using laser scanning confocal microscope Olympus FV1200 (Germany). The mean fluorescence intensity was measured with ImageJ software using “measure tool” [38]. Fluorescence intensity values from anteroventral thalamic nucleus were considered as “1,0” and intensity values from images containing anteroventral thalamic nucleus, internal medullary lamina of the thalamus, mediodorsal thalamic nucleus magnocellular part, mediodorsal thalamic nucleus parvocellular part, anterior ventrolateral nucleus and substantia nigra pars reticulata were normalized to fluorescence intensity of anteroventral thalamic nucleus (Figure 1F). Staining specificity and representative figures from AUD brain are shown in Appendix A.

## 3. Results

### 3.1. Differential Gene Expression

We analyzed thalamus tissues for the molecular changes induced by chronic alcohol use. First, we performed a routine data quality control according to the TAC software user guide. The data for samples of two subjects (one AUD and one control) did not pass the Hybridization Controls Threshold, and thus were excluded from the further data analysis. This left us with 5 samples in the AUD group and 4 samples in the control group, which we could use for the following differential gene expression analysis.

We found statistically significant (FDR ≤ 0.05) alcohol induced changes in gene expression (Appendix A). Altogether 4695 genes were differentially expressed in thalamus tissue of AUD subjects compared to controls. Out of all these genes, 2802 were up-regulated and 1893 were down-regulated.

We clustered the differentially expressed genes applying Heatmap analysis. However, here we used more stringent conditions to see the effect of most significantly affected genes—FDR ≤ 0.01 and Fold Change ≥ ±2. (Figure 2). Thus altogether 931 DEGs were included.

### 3.2. Functional Pathway Analysis

The 931 DEGs used for Heatmap clustering analysis were used as input in the Gene Ontology analysis applying the g:GOSt tool. The tool was able to use 914 genes and resulted in 313 statistically significant findings (adjusted-*p* value between 1.16 × 10^−21^ and 0.049) listed in the Appendix A.

The submission of the full list of significantly upregulated DEGs (FDR ≤ 0.05; 2802 genes) to the Gene Ontology GO enrichment analysis tool revealed 53 GO “molecular function” annotation categories (FDR *p*-value < 1 × 10^−2^ and enrichment fold ≥ 2), whereas the submission of down-regulated DEGs (1893 genes) revealed five enriched GO molecular function categories according the same criteria of significance (Appendix A).

The WikiPathway analysis via TAC resulted in a list of 620 pathways from which 69 pathways had *p*-value ≤ 0.05 and included ≤2 DEGs (listed in the Appendix A); 13 pathways from these included ≤10 DEGs. The schematic images representing Wikipathway functional pathway analysis output (FDR *p* < 0.01) can be found in Appendix A.

Most of the genes with significantly altered expression are related to the functional categories related to the general neurotransmission activity, such as “synapse” (GO:0045202, *p* = 1.16 × 10^−21^), “synaptic signaling” (GO:0007268, *p* = 2.04 × 10^−14^), “trans-synaptic signaling” (GO:0099537, *p* = 2.99 × 10^−13^). (Appendix A).

Most of the significantly upregulated gene clusters (FDR < 0.05) are related to neuronal activity and metabolism, namely the functional categories “TCA Cycle (aka Krebs or citric acid cycle)” (Wikipathway, *p* = 0.001, Appendix A) and “Fatty Acid Biosynthesis” (Wikipathway, *p* < 0.001, Appendix A) included only upregulated genes. A variety of neurotransmitter receptors, mostly glutamatergic and GABAergic receptors are represented in functional categories, enriched among upregulated genes, such as: “glutamate receptor binding” (GO:0035254, *p* = 1.70 × 10^−4^), also “glutamatergic synapse” (GO:0098978, *p* = 1.90 × 10^−10^); “regulation of glutamate receptor signaling pathway” (GO:1900449, *p* = 0.0004); “GABA receptor binding” (GO:0050811, *p* = 2.71 × 10^−2^), “GABAergic synapse” (KEGG:04727, *p* = 2.88 × 10^−5^). Significantly altered gene cluster “Dopaminergic synapse” (KEGG:04728, *p* < 0.05) includes 12 upregulated versus 2 downregulated genes. Significantly altered pathway “Corticotropin-releasing hormone signaling pathway” (Wikipathway, *p* < 0.05) includes 9 upregulated genes versus 1 downregulated gene. Significantly down-regulated functional gene clusters were related to olfactory pathways: “odorant binding” (GO:0005549, *p* = 3.59 × 10^−5^), “olfactory receptor activity” (GO:0004984, *p* = 9.85 × 10^−20^) or cytokine signaling: “cytokine activity” (GO:0005125, *p* = 5.68 × 10^−3^), “cytokine receptor binding” (GO:0005126, *p* = 3.64 × 10^−3^). Significantly altered gene cluster “Chemokine Signaling Pathway” (FDR *p* = 0.01, Appendix A) also reveals the prevailing downregulation of cytokines. For further details see Appendix A and Appendix A.

By using WikiPathway analysis tool, we depicted selected pathways from the list of 69 significantly enriched gene clusters to illustrate the alternated functional networks in the brains of AUD subjects: “Common Pathways Underlying Drug dependence” (FDR *p* < 0.01, Figure 3), “TCA Cycle (aka Krebs or citric acid cycle)” (FDR *p* = 0.001, Appendix A); “Fatty Acid Biosynthesis” (FDR *p* < 0.001, Appendix A). “GABA receptor Signaling” (FDR *p* = 0.001; Appendix A), “Synaptic Vesicle Pathway” (FDR *p* < 0.001, Appendix A); “Disruption of postsynaptic signaling by CNV” (FDR *p* = 0.0001; Appendix A), “G Protein Signaling Pathways” (FDR *p* = 0.0001, Appendix A), “Chemokine Signaling Pathway” (FDR *p* = 0.01, Appendix A) and “Cannabinoid receptor signaling” (FDR *p* < 0.05, Appendix A).

### 3.3. Gene Expression Changes in the Context of Recent GWAS Studies

The list of significantly altered genes in the dorsomedial thalamus of the AUD subjects was compared to the list of genes, significantly associated with AUD and alcohol consumption phenotypes in recent (2019–2020) GWAS studies. From 42 genes that were shown to have strong associations (FDR *p* < 10^−8^) with alcohol-related phenotype, 30 were also significantly (FDR *p* < 0.05) upregulated in the dorsomedial thalamus of AUD subjects and 12 genes were significantly downregulated. The significantly altered genes from the current study that were confirmed as significant in at least two GWAS studies, are listed in Table 1, and the full list of 42 overlapping genes is in the Appendix A. Interestingly, most of the genes supported by both gene array and GWAS data were associated with both AUD and alcohol consumption phenotypes, such as DRD2, FTO, TCF4, CADM2, CRHR1, synaptotagmin 14, ADH1B and GCKR (Appendix A).

### 3.4. Western Blot

In order to provide an independent validation of transcriptional changes seen in the microarray, we performed western blot analysis. We confirmed the upregulation of FTO and GRIN1 in the dorsolateral thalamus of AUD subjects both in the transcript and protein level (Figure 4A,B). AUD subjects had elevated but not statistically significant increase of CYT-C protein abundance concurrent with the upregulation of CYT-C in transcriptional level (Figure 4C). It is important to note that transcript levels and protein levels are not always highly correlated therefore quantitative qRT-PCR would have been the preferred method for chip assay validation. However, supported by earlier studies showing generally high correlations of mRNA-to-protein abundance [41], we can assume that if we can see significant rise of both GRIN1 and FTO, corresponding transcripts are also likely upregulated. No change in the expression of GAPDH protein was observed on Western blot (Figure 4D) unlike microarray analysis.

### 3.5. Immunochemistry

In order to describe GRIN1 localization in human thalamus, we used indirect immunolabeling against human GRIN1 protein. Immunofluorescence staining revealed localization of GRIN1 as puncta (presumably synapses) in all observed regions (Figure 1B–E,G,H). In addition, in some of the cases we can observe GRIN1 positive puncta forming continuum in thalamic nuclei (arrows) suggesting several glutamatergic inputs on the different levels of the same process. In order to quantify gathered results, we measured fluorescence intensity values from images containing anteroventral thalamic nucleus, internal medullary lamina of the thalamus, mediodorsal thalamic nucleus magnocellular part, mediodorsal thalamic nucleus parvocellular part, anterior ventrolateral nucleus and substantia nigra pars reticulata. The intensity values were normalized to anteroventral thalamic nucleus (considered 1.0) and the results are given in Figure 1F.

## 4. Discussion

### 4.1. The Imbalance between Neuronal Excitation and Inhibition

Dorsomedial thalamus is an important subcortical relay with well described impact in the formation and progress of alcohol induced neuronal damage, still, diencephalon, including thalamic nuclei have been underrepresented in the high-throughput studies of chronic alcohol induced molecular changes in the human brain. We detected strong alterations in the well-known direct targets of ethanol [42], such as gene clusters related to the GABAergic, glutamatergic, dopaminergic, endocannabinoid-mediated and corticotropin releasing factor (*CRF*)-related neurotransmission, but also provided evidence about the extended profile of alterations in addition to well characterized molecular targets, many of them also overlapped with the significant findings from recent GWAS studies for alcohol dependence or related phenotypes.

Chronic exposure to heavy alcohol consumption is known to result in the imbalance in neuronal excitation and inhibition, caused by widespread neuronal alterations in transmission of excitatory glutamate and inhibitory gamma-aminobutyric acid (GABA). [36,43]. It has been shown that acute ethanol exposure magnifies the effects of GABA and inhibits the function of NMDA receptor (NMDAR) [42] whereas long-term ethanol exposure generates compensatory downregulation of GABA and GABA receptors and upregulation of NMDA receptors, resulting in increase in glutamatergic activity [44,45], therefore chronic alcohol use results in a pronounced hyperglutamatergic/hyperexcitable state of the central nervous system [46]. In the current study, we showed the upregulation of the genes related to both glutamatergic (Figure 3) and also GABAergic neurotransmission (Appendix A) in the dorsomedial thalamus of AUD subjects. The increase in the glutamatergic neurotransmission in the brains of AUD subjects is expected according to the earlier knowledge and we showed the upregulation of GRIN1 in the dorsolateral thalamus of AUD subjects both in the transcript and protein level (Figure 4B). The increase in synaptic vesicle pathway (Appendix A) could be also related with the hyperexitatory condition in the brains of AUD subjects. The increase of GRIN1 in the brains of people with chronic alcohol use was previously shown in the hippocampus but not in the frontal cortex [19] suggesting, together with the current findings, that hyper-glutaminergic state can be more prevalent in the subcortical structures. Immunohistochemical staining revealed a similar heterogenous distribution of GRIN1 in the thalamic nuclei of both AUD and control subjects indicating variable functional importance in distinct thalamic nuclei.

The abundant upregulation of the genes encoding proteins that are central in the GABAergic neurotransmission, could be also compensatory. For example, the upregulation of GAD1 and GAD2 transcripts that are encoding glutamate decarboxylase isoforms, which are catalyzing the synthesis of GABA from glutamate, could be compensatory in the conditions of low GABA that has been described in the brains of AUD subjects. It is important to note that Enoch et al. [47] showed a decrease of GABA receptor 2A (GABRA2) in the hippocampi of AUD subjects, whereas they did not find changes in the transcripts encoding GAD enzymes, whereas GAD1 transcript and gephryin were upregulated in the hippocampi of alcohol preferring rats, coinciding with our current results in the dorsomedial thalamus of human subjects. Further studies are needed to specify if the alterations of genes and proteins for GABAergic neurotransmission are specific to certain brain areas in AUD subjects.

### 4.2. Altered Transcripts Encoding Genes Related to Cellular Metabolism

We are currently showing an abundant increase of transcripts encoding genes related to cellular metabolism in the dorsomedial thalamus of AUD subjects. The genes encoding enzymes participating in the tricarboxylic acid (TCA) cycle or the Krebs cycle are upregulated (Appendix A) in the AUD subjects as well genes related to fatty acid synthesis (Appendix A). We propose that the hyperglutamatergic and hyperexcitable state of the cells in the brains of AUD subjects [46] also raises the need for more cellular energy that is needed for the increased excitatory neurotransmission. This upregulation of the transcripts of genes encoding energy-generating proteins can, again, also be compensatory, as recent proteomic analysis that found several alterations in the central brain metabolism in the cortex areas of AUD brains, did not found significant changes in abundance of the enzymes of the TCA cycle [48].

In the current study, the significand upregulation of transcript encoding glyceraldehyde-3-phosphate dehydrogenase (GAPDH) in the dorsomedial thalamus of AUD subjects, lead us to study if the levels of GAPDH protein is also altered. The protein levels of GAPDH were highly variable in the AUD study group (Figure 4D), therefore, no significant alterations were detected. As GAPDH is also among the enzymes controlling this energy production pathway, we can suggest that since metabolic alterations in the brains of subjects with chronic alcohol use have been repeatedly described [45,48], GAPDH is not a suitable housekeeping gene in the studies that measure gene or protein expression alterations in the brains of subjects with a history of chronic alcohol use. Indeed, growing evidence suggests that GAPDH, initially identified as a glycolytic enzyme and considered as a housekeeping gene, is actually involved in numerous cellular functions [49]. At the same time, in the current study, the significant upregulation of cytochrome C encoding transcript (CYCS) was supported by the tendency level increase of the corresponding protein (Figure 4C), however, more research is needed to clarify the evidence of altered metabolism in the brains of AUD subjects.

### 4.3. Thalamic Gene Expression Changes in the Context of Recent GWAS Studies

Several recent GWAS studies representing analyses of over 100,000 study subjects have provided known and novel loci significantly associated with the traits related with chronic alcohol use. Few of these loci encode well known direct targets of alcohol, such as alcohol dehydrogenase 1B (ADH1B), however, the causality and function of the significant loci often remains elusive, due to incomplete knowledge of molecular functions. To expand the biological insight of the GWAS results, we compared our results with genes identified in recent genome-wide association studies (GWAS) for AUD or related phenotypes. From 16 genes that have been shown to have strong associations with AUD or alcohol consumption phenotypes in at least two independent GWAS studies, 12 genes were also significantly upregulated in the dorsomedial thalamus of AUD subjects and 4 genes were significantly downregulated (Table 1). As the expression of alcohol dehydrogenase subunits is minimal in the brain, it is surprising that three alcohol dehydrogenase subunits (ADH1A, ADH1B and ADH7) were significantly downregulated in the dorsolateral thalamus of the subjects with AUD. However, it has been shown that for example *ADH1B* has complex mechanisms that regulate its expression across multiple human tissues which may be involved in multiple molecular pathways [50].

Dopamine receptor 2 (DRD2) and FTO were two genes that have been repeatedly shown to be associated with AUD and alcohol consumption phenotypes in GWAS studies; also, the transcripts encoding these proteins were significantly upregulated in the dorsomedial thalamus of AUD subjects and the upregulation of FTO was also confirmed in the protein level. The genomic locus of FTO gene, also known as alpha-ketoglutarate dependent dioxygenase, has been strongly associated with BMI and obesity [51]. Besides significant associations with AUD and alcohol consumption phenotypes [4,5,8], FTO also appears to be one of the most important loci associated with food intake [52] suggesting that FTO gene could be a pleiotropic locus that has impact on common reward pathways implicated in the excess consumption of both food and alcohol. Indeed, it has been shown that FTO participates in the regulation of D2R-dependent reward learning and modulates the connectivity in a basic reward circuit of meso-striato-prefrontal regions [53]. Furthermore, a significant association between polymorphisms in the FTO gene and alcohol consumption remained in case the analysis was adjusted for BMI, suggesting that the associations for alcohol consumption are independent of BMI, supporting the hypothesis of a potentially shared, rather than mediated, pathways [9]. The significantly altered functional pathway “Common Pathways Underlying Drug dependence”, depicted in Figure 3, illustrates the mechanisms through which dopamine and glutamate neuroactive ligand-receptor interactions trigger long-term potentiation underlying the development of addiction [40].

Several significantly associated genes from alcohol consumption related GWAs, being also significantly upregulated in the thalamus of AUD subjects, are involved in the neurodevelopment and/or of neurodegenerative brain diseases. Transcription factor TCF4 binding sites are found in a large number of neuronal genes which makes TCF4 an important regulator of neurodevelopment and neuronal survival [54]. DPP6 plays a crucial role in neuronal excitability and is also associated with dementia [55]. CADM2 is a neural cell adhesion molecule (CAM) that maintains neuronal integrity regulating cognitive processing speed [56]. Furthermore, numerous other neural CAMs are upregulated along with CADM2, such as NRXN1, NRXN2, NRXN3, NCAM1, NRCAM, L1CAM, CADM1, CADM2, CADM3, CADM4 and IgLON family adhesion molecules: LSAMP, NTM and OPCML, that are also involved in the development of neuropsychiatric disorders [57]. In conclusion, the upregulation of numerous genes that are involved in neuronal integrity and survival emphasizes the strong overlap between aspects of AUD and other psychiatric disorders [58].

### 4.4. Downregulation of Immune Response Related Genes

Not many functional categories were significantly downregulated in the current study, according to GO enrichment analysis the functional categories were related to either olfactory signaling (“olfactory receptors” and “odorant binding”) or cytokine signaling (“cytokine activity” and “cytokine receptor binding”). Interestingly, in addition to the canonical signaling pathways activated by olfactory receptors (ORs) in olfactory sensory neurons, alternative pathways have been demonstrated in numerous non-olfactory tissues including brain and immune cells [59]. Kerslake et al. [60] have recently shown the co-expression of peripheral olfactory receptors with SARS-CoV-2, however, further studies are needed to confirm if the downregulation of the genes encoding olfactory receptors could be related to the downregulation of the genes encoding cytokine signaling. It has been well described that chronic alcohol exposure disrupts the balance between cell types and signal molecules in the immune system [61]. We found that the majority of altered cytokines of the chemokine family (CCL1, CCL7, CCL15, CCL17 PF4, CXCL1, CXCL2) and their receptors (CXCR5, CXCR6) were downregulated in the transcript level in the dorsomedial thalamus of AUD subjects, whereas most of the genes encoding downstream elements for cytokine signaling (g-protein related signaling) were upregulated (Appendix A). It is important to note that CXCL2 transcript has been shown to have PMI-time dependent fluctuating expression profile in several tissues [62] and therefore the downregulation of CXCL2 must be interpreted with caution.

Numerous other significantly down-regulated transcripts encode well known factors in the immune system, such as interferons; interferon gamma (IFNG) and several others: IFNA2, IFNA10, IFNA17, IFNA21, IFNL2, and IFNL3. Furthermore, all altered transcripts encoding interleukins (IL1F10, IL2, IL17A, IL17F, IL21, IL23A, IL31, IL36B, IL36G, IL37) and interleukin receptors (IL1R1, IL2RB, IL22RA2, IL27RA) are also exclusively downregulated in the dorsomedial thalamus of AUD subjects, indicating the systemic downregulation of immune mediators/receptors in the thalamus of AUD brain.

## 5. Conclusions

We detected strong alterations in the well-known direct targets of ethanol, such as gene clusters related to the GABAergic, glutamatergic and dopaminergic neurotransmission, but also provided insight into how significant AUD and alcohol consumption related GWAS results relate to the molecular network affected by chronic alcohol use in the thalamus. Molecular targets that are supported by both microarray and GWAS data deserve further focused research in the future studies of the genetics of AUD and alcohol consumption phenotypes.

The present study is small, but we were able to detect changes in the thalamus caused by alcohol indicating that the alterations in the thalamus are robust. Furthermore, our study extended the understanding of the associations gained in recent significant GWAS associations. Our results emphasize the importance of deeper brain structures, such as diencephalon in the development of chronic alcohol use-related neuroadaptations which are important to acknowledge in order to understand the neurobiological mechanisms of addiction and withdrawal.

## Figures and Tables

**Figure 1 brainsci-11-00435-f001:**
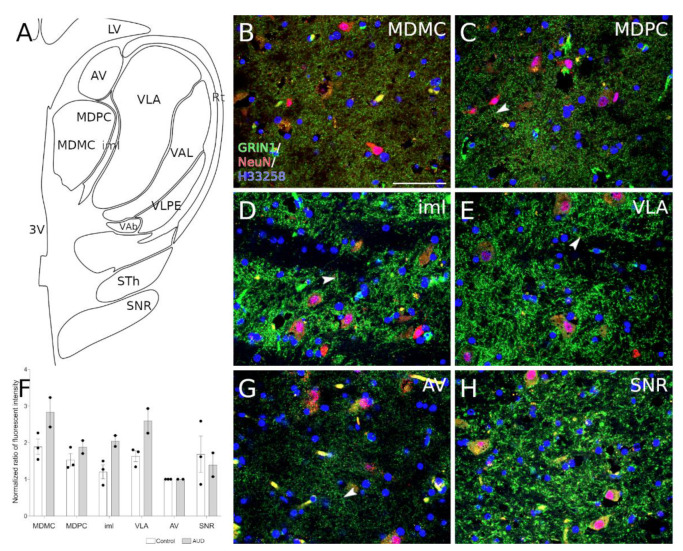
GRIN1 expression in the human thalamic region. (**A**) schematic drawing from−16.83 MNI displaying thalamic nuclei. (**B**–**E**,**G**,**H**) representative images show immunohistochemical staining of GRIN1 (green) and neuronal nuclei (red) from selected thalamic nuclei. Localization of GRIN1 protein is observable as puncta. (**C**,**D**,**E**,**H**) in the nuclei where the expression of GRIN1 is strongest we observe periodic localization of GRIN1 (arrowheads). Cell nuclei were stained with H33258 (blue). (**F**) graph shows the average normalized ratio of fluorescence intensity from analyzed structures to anteroventral thalamic nucleus (see section in material and methods). Abbreviations: 3V, third ventricle; AV, anteroventral thalamic nucleus; iml, internal medullary lamina of the thalamus; LV, lateral ventricle; MDMC, mediodorsal thalamic nucleus, magnocellular part; MDPC, mediodorsal thalamic nucleus, parvicellular part; Rt, reticular thalamic nucleus; SNR, substantia nigra pars reticulata; STh, subthalamic nuclei; VAb, basal ventroanterior nucleus; VAL, lateral ventroanterior nucleus; VLA, anterior ventrolateral nucleus; VLPE, posterior ventrolateral nucleus, external part. Scalebar: 50 μm.

**Figure 2 brainsci-11-00435-f002:**
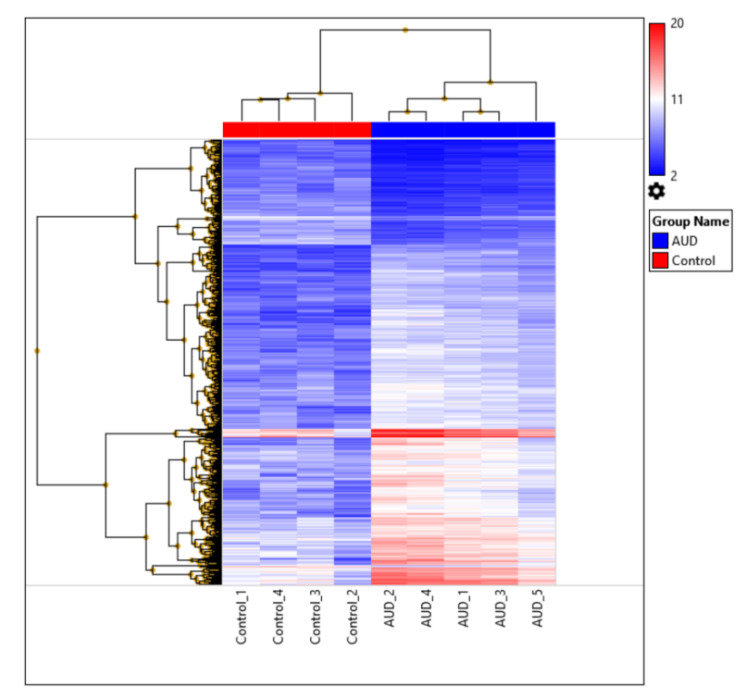
Heatmap of the 931 most significantly differentially expressed genes in the thalamus (FDR < 0.01 and Fold Change ≥ ±2). Clear distinction between the alcohol use disorder (AUD) group and controls is evident by the gene expression profiles in the dorsomedial thalamus. The legend represents fold change based on linear data.

**Figure 3 brainsci-11-00435-f003:**
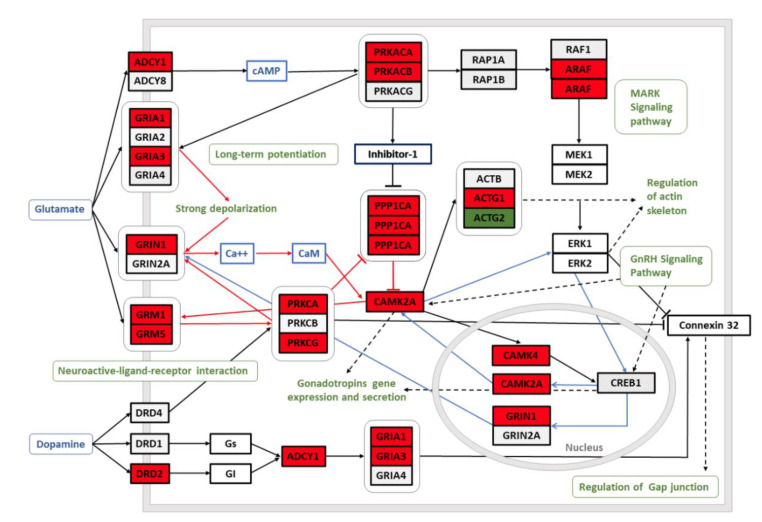
The schematic model of “Common Pathways Underlying Drug dependence” is an output (FDR *p* < 0.01) of the Wikipathway functional pathway analysis [39]. Red background of the genes indicates significant upregulation and green background downregulation of the gene in the dorsomedial thalamus of the AUD subjects. The schema is originally constructed by Li et al. [40]: glutamate and dopamine neuroactive ligand-receptor interactions trigger long-term potentiation, MAPK and GnRH signaling and gap junction regulation.

**Figure 4 brainsci-11-00435-f004:**
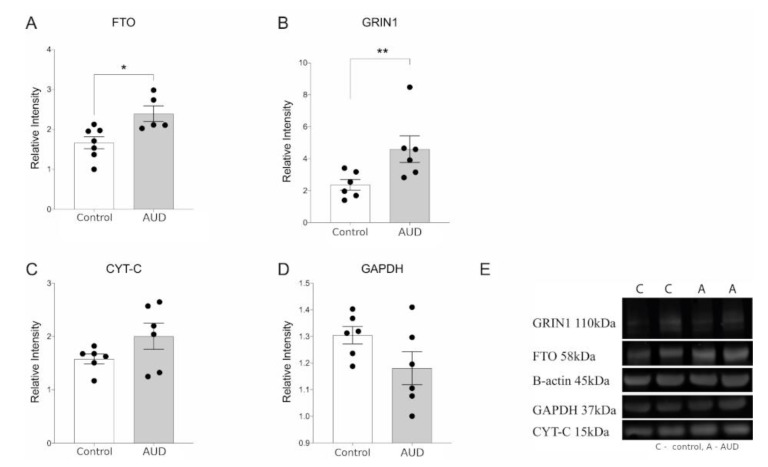
Protein expression in thalamus of AUD and control subjects—(**A**) FTO–FTO Alpha-Ketoglutarate Dependent Dioxygenase, (**B**) GRIN1—glutamate ionotropic receptor NMDA type subunit 1, (**C**) Cyt-C—Cytochrome c, (**D**) GAPDH—Glyceraldehyde 3-phosphate dehydrogenase, and (**E**) representative immunoblots. Data are expressed as mean values ± SEM (*n* = 5–6). Statistical analysis was performed with unpaired t-test with Welch’s correction. * *p* ≤ 0.05, ** *p* ≤ 0.01 when compared with controls.

**Table 1 brainsci-11-00435-t001:** 16 genes that were differentially expressed (FDR *p* < 0.05) in the dorsomedial thalamus of AUD subjects and have also been found to be significantly (FDR *p* < 10^−8^) associated with AUD or alcohol consumption phenotypes in at least two independent recent GWAS studies (2019–2000: Evangelou et al., 2019 [4]; Liu et al., 2019 [6]; Sanchez-Roige et al., 2019 [7]; Zhou et al., 2020 [8]; Thompson et al., 2020 [9]; Gelernter et al., 2019 [34]; Karlsson et al., 2019 [35]). The highest GWAS *p*-value has been given. More detailed information about GWAS phenotypes and *p*-values can be found in Appendix A.

Gene Symbol	Gene Alias/Full Name	Gene Array Fold Change	GWAS Link with Alcohol Consumption Phenotype	Highest GWAS *p* Value
ARID4A	AT rich interactive domain 4A	4.43	[6,8]	1 × 10^−8^
ARPC1B	Actin related protein 2/3 complex subunit 1B	3.08	[4,6]	3 × 10^−10^
CADM2	Cell adhesion molecule 2	2.87	[4,6,8,35]	2 × 10^−17^
CRHR1	Corticotropin releasing hormone receptor 1	6.27	[7,34]	1.63 × 10^−9^
DPP6	Dipeptidyl-peptidase 6	2.56	[4,6,8,35]	6 × 10^−14^
DRD2	Dopamine receptor D2	5.22	[5,6,8,9,35]	5.13 × 10^−15^
FTO	Alpha-ketoglutarate dependent dioxygenase	7.75	[4,5,8]	1.11 × 10^−19^
MAPT	Microtubule associated protein tau	5.2	[4,7]	5 × 10^−23^
OTX2	orthodenticle homeobox 2	2.65	[4,6]	2 × 10^−12^
POLR3H	RNA polymerase III subunit H	7.09	[6,35]	5 × 10^−9^
SEZ6L2	Seizure related 6 homolog like 2	8.25	[4,6,35]	3 × 10^−15^
SORL1	Sortilin-related receptor 1	3.46	[6,8]	2.24 × 10^−9^
SYT14	Synaptotagmin 14	2.68	[4,8]	1 × 10^−10^
TCF4	Transcription factor 4	5.39	[4,6,8,35]	1 × 10^−10^
ADH1B	alcohol dehydrogenase 1B	−3.54	[4,5,6,8,9,34,35]	2.2 × 10^−308^
ADH7	alcohol dehydrogenase 7	−2.48	[6,35]	2 × 10^−22^
EMCN	Endomucin	−3.2	[6,35]	3 × 10^−23^
GCKR	glucokinase regulator	−4.7	[4,5,6,7,8,9,35]	2 × 10^−60^
SIX3	SIX homeobox 3	−2.82	[5,8]	2.26 × 10^−15^

## Data Availability

The data presented in this study are available on request from the corresponding author.

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
