# Peer review of "Chronic Alcohol Use Induces Molecular Genetic Changes in the Dorsomedial Thalamus of People with Alcohol-Related Disorders"

_brainsci, 2021, doi:10.3390/brainsci11040435_

Round 1
Reviewer 1 Report
The manuscript is well-structured and provide interesting new data regarding the changes in gene expression in the mediodorsal thalamus nucleus of the human brain. Postmortem human brain studies are important in this field but caution is needed to reduce influence by postmortem artifacts. The authors recorded pH in the lateral ventricle; the brain pH is most commonly measured in homogenates of cerebellum, did they also record that? They did exclude cases with too low pH values, did they also exclude some cases due to poor RIN values? The postmortem interval should be divided into warm time and cold time since the length of the warm time is the most critical period. It should not be a problem for the authors to specify this, since it is highly likely that the arrival of the bodies to the morgue has been recorded. The determination of alcohol use from alcohol-related changes in the liver, heart and pancreas is different from what is typically used. It is unclear why they report that they have excluded non-alcoholic causes of liver cirrhosis by checking medical records whereas they do not mention the use of medical record to support alcohol use disorder. Pancreatitis is not specific to alcohol. The authors should specify the features of alcoholic cardiomyopathy that they used for inclusion and provide a reference. They performed gene chip analysis on 11 cases and Western on 12 cases, but both analyses on only one case, why didn't they perform both genechip and Western on all cases? Did they have preliminary data to perform a power analysis before launching the study? Did they cross-check the changes in gene expression for select genes that have been reported to occur during agony/supravital period (data from the GTEx project, see e.g. Ferreira et al Nature comm 2018;9:490)? Interestingly, the authors report highly variable GAPDH protein levels in alcoholics that actually show an upregulation of the GADPH gene, could the protein levels be correlated to other. Did the authors look for other markers of altered metabolism (they mention CYCS, but state it was at "tendency level"). Given that several markers may be associated with altered metabolism, was there any of the alcohol subjects that suffered from alcohol ketoacidosis (or even died by this condition) prior to their demise? (and what about the controls, did any of them show signs of ketoacidosis of any kind)? Were subjects with diabetes included?
Author Response
Response to Reviewer 1 Comments:
Question 1.1. The authors recorded pH in the lateral ventricle; the brain pH is most commonly measured in homogenates of cerebellum, did they also record that? They did exclude cases with too low pH values, did they also exclude some cases due to poor RIN values? The postmortem interval should be divided into warm time and cold time since the length of the warm time is the most critical period. It should not be a problem for the authors to specify this, since it is highly likely that the arrival of the bodies to the morgue has been recorded.
Answer 1.1. We thank the reviewer to point out an important issues wit pH and PMI. The tissue pH was initially measured routinely in several brain locations including cerebellum. As the measurements were generally consistent with the values received from lateral ventricles, we later chose measurement only from lateral ventricles as according to our observations it was generally well representing the pH values in other brain areas. No cases were excluded due to RIN. The post mortem interval has now been divided into warm time and included to the Supplementary table S1. Also, a new section has been added to the “2.Materials and methods” part (page 3 section 2): „Median warm time (time before cold storage at 4 °C) was 5.61±3.13 h for alcoholics and 6.75±2.97  h for control subjects, therefore most of the PMI represents cold time, which slows down postmortem tissue degradation (Dhanabalan et al, 2018).”
Question 1.2. The determination of alcohol use from alcohol-related changes in the liver, heart and pancreas is different from what is typically used. It is unclear why they report that they have excluded non-alcoholic causes of liver cirrhosis by checking medical records whereas they do not mention the use of medical record to support alcohol use disorder. Pancreatitis is not specific to alcohol. The authors should specify the features of alcoholic cardiomyopathy that they used for inclusion and provide a reference.
Answer 1.2. We agree that these aspects need to be clarified. In all cases, cause of death was determined based on previous medical records, autopsy signs and routine laboratory analysis. In present study we only included subjects with prior clinical diagnosis of alcoholic cardiomyopathy and pancreatitis in medical records, supported by histology taken from tissue samples during the autopsy. If supportive medical history would not be available, the diagnosis would be based on left ventricular dilatation in presence of alcoholic liver disease (ALD) supported by histology indicating characteristics previously described in this disorder (Piano and Phillips, 2014; Guzzo-Merello, Cobo-Marcos, Gallego-Delgado, and Garcia-Pavia 2014; Fernández-Solà 2020).
We have accordingly modified page 2 and 3 paragraph 1 to emphasize this point:
“Chronic alcohol abuse of the deceased (n=15) was based on prior diagnosis of alcohol related disorder together with clinical diagnosis of alcohol related end-organ damage supported by post-mortem histology, including alcoholic liver disease, alcoholic pancreatitis, and alcoholic cardiomyopathy. Non-alcoholic study group (n=12) included subjects which had no prior diagnosis of alcohol related disorder code, end-organ damage, and signs of alcohol related damage collected from tissue samples taken during the autopsy. The subjects with liver cirrhosis were only included into the control group in case of confirmed clinical diagnosis of non-alcoholic liver cirrhosis (due to biliary cirrhosis or hepatitis C).”
Piano, M.R., and Phillips, S.A. (2014). Alcoholic cardiomyopathy: pathophysiologic insights. Cardiovasc. Toxicol. 14(4):291-308. doi:10.1007/s12012-014-9252-4
Guzzo-Merello, G., Cobo-Marcos, M., Gallego-Delgado, M., and Garcia-Pavia, P. (2014). Alcoholic cardiomyopathy. World journal of cardiology. 6(8), 771–781. https://doi.org/10.4330/wjc.v6.i8.771
Fernández-Solà J. (2020). The Effects of Ethanol on the Heart: Alcoholic Cardiomyopathy. Nutrients. 12(2), 572. https://doi.org/10.3390/nu12020572
Question 1.3. They performed gene chip analysis on 11 cases and Western on 12 cases, but both analyses on only one case, why didn't they perform both genechip and Western on all cases? Did they have preliminary data to perform a power analysis before launching the study?
Answer 1.3. We agree that it would be useful to see the microarray scores and western blot scores in the same individuals. Currently we choose different study subjects for western blot in order to confirm the same molecular changes in an independent study group and in order to provide evidence of the significance of microarray data in the protein level. The explanation has now been improved in the “Materials and methods” section (page 3, paragraph 2).
Due to the limited access to post-mortem tissues, we were not able to perform a pilot study evaluating the potential amount of variation in the data, which is needed for the power calculations. Also the minimal effect size, which would be clinically or scientifically meaningful is hard to evaluate at the current knowledge stage. Thus, with a small sample available for us we expected to see an occurrence of medium or large effects in the thalamic area of alcoholic subjects versus controls. The section pointing out this methodological limitation has now been added to the “Materials and methods” section (page 4, paragraph 2), the statements that the study is “sufficiently powerful” have now been removed from the “Abstract” and “Conclusions”
Question 1.4. Did they cross-check the changes in gene expression for select genes that have been reported to occur during agony/supravital period (data from the GTEx project, see e.g. Ferreira et al Nature comm 2018;9:490)? Interestingly, the authors report highly variable GAPDH protein levels in alcoholics that actually show an upregulation of the GADPH gene, could the protein levels be correlated to other. Did the authors look for other markers of altered metabolism (they mention CYCS, but state it was at "tendency level").
Answer 1.4. We thank the reviewer to point out an important issue about transcripts related with PMI periood. We have now cross-checked the altered gene list with the genes that have been shown to be significantly altered during the post mortem period (Ferreira et al, 2018). Indeed, CXCL2 that is downregulated in the thalamic area of alcoholic subjects along with other cytokines of the chemokine family has been shown to have PMI-time dependent fluctuating expression profile in several tissues (Ferreira et al, 2018). Although CXCL2 has not been shown to change reactively in the brain tissues, we agree that the downregulation of CXCL2 has to be interpreted with caution. This notion has now been added to the “Discussion” section (end of page 14 – page 15, first paragraph). In the current study we did not look other markers of altered metabolism. We agree that more research is needed to clarify the evidence of altered metabolism in the brains of alcoholic subjects. This implication has now also been added to the “Discussion” section (page 13, paragraph 2)
Question 1.5. Given that several markers may be associated with altered metabolism, was there any of the alcohol subjects that suffered from alcohol ketoacidosis (or even died by this condition) prior to their demise? (and what about the controls, did any of them show signs of ketoacidosis of any kind)? Were subjects with diabetes included?
Answer 1.5. Thank you for pointing this out. We agree that given the importance of ketoacidosis in alcoholics, we should have exemplified it as well. Subjects with a known diagnosis of diabetes and/or prior history of ketoacidosis, including diabetic and/or alcoholic, were excluded. Toxicological analysis was routinely performed and none of the selected subjects presented acetone levels indicating ketoacidosis, in suspected cases additional biochemical analysis of beta-hydroxybutyrate (BHB) would have been done.
We have accordingly modified page 3 paragraph 1, to emphasize these aspects:
Subjects with a known diagnosis of diabetes and/or prior history of ketoacidosis, including diabetic and/or alcoholic, were excluded. Toxicological analysis was routinely performed and none of the selected subjects presented acetone levels indicating ketoacidosis.
Reviewer 2 Report
In this manuscript, authors have performed microarray assay and have identified differentially expressed genes in dorsomedial thalamus from human postmortem brain (chronic alcoholics=15 and control=12). Most of those altered gene were associated with glutamatergic signaling, GABAergic signaling, cellular metabolism and neurodevelopment. They have also found that most of the DEGs in this study are associated with recent GWAS. Although there is no clear hypothesis in this study, this finding from dorsomedial thalamus would help in future AUD studies.
I have few comments/concerns:
- It would have been nice if had used same subjects for microarray and western blot exp. So can clarify the reason in the text. Also the way it is written (page 3, paragraph 2), it is confusing.
- Western blot won’t validate the transcript data as messenger RNA not necessarily correlate with the protein translation. So, it would have been nice if they had done qRT-PCR of GRIN1 and other highly altered genes using the same sample they used in Chip assay to validate the microarray result. Since western and IHC not helping much to validate the microarray results.
- The result section for Immunohistochemistry data is missing. Have not mentioned about this.
- Regarding the normalization of GluN1 IHC, is florescent value of anteroventral thalamic nucleus of both groups same? If that’s different then whole results will be affected. Also, are those experiments done all at once? Otherwise background florescence is different. Also, since it just has 2 subjects in alcoholic group, wont be reliable. It is not clear why authors could not have n=3 for this group. This data really not adding much besides showing an IHC of GluN1 which is also not great (background to signal noise is high). Representative western image is also not convincing if you look at grin1 blot.
- In Table 1, it would be helpful if authors include fold change of the gene in this study they are associated with GWAS studies.
- In Figure 3, it will be helpful for readers if they write down antibody with the color code on the top of image. Also, it is better to have a scale bar. Also figure legend is not clear: is it from control or alcholoic brain? It will be better if authors could put images from both subjects side by side for visual comparison.
- Also, I think it is helpful to have heat map image of highly altered DEGs on main manuscript or have a table of some of the highly altered DEGs so readers can look at it easily.
Author Response
Response to Reviewer 2 Comments:
Question 2.1. It would have been nice if had used same subjects for microarray and western blot exp. So can clarify the reason in the text. Also the way it is written (page 3, paragraph 2), it is confusing.
Answer 2.1. We agree that it would be useful to see the microarray scores and western blot scores in the same individuals. Currently we choose different study subjects for western blot in order to confirm the same molecular changes in an independent study group and in order to provide evidence of the significance of microarray data in the protein level. The explanation has now been improved in the “Materials and methods” section (page 3 paragraph 2).
Question 2.2. Western blot won’t validate the transcript data as messenger RNA not necessarily correlate with the protein translation. So, it would have been nice if they had done qRT-PCR of GRIN1 and other highly altered genes using the same sample they used in Chip assay to validate the microarray result. Since western and IHC not helping much to validate the microarray results.
Answer 2.2. It is true that transcript levels and protein levels are not always highly correlated. However, supported by earlier studies showing generally high correlations of mRNA-to-protein abundance (Magnusson et al, 2020) we could assume that if we can see significant rise of both GRIN1 and FTO, corresponding transcripts are also likely upregulated. Corresponding explanation has now been added to the “Results” section (page 11 paragraph 1)
Magnusson R, Rundquist O, Kim MJ, Hellberg S, Na CH, Benson M et al (2020) A validated strategy to infer protein biomarkers from RNA-Seq by combining multiple mRNA splice variants and time-delay. bioRxiv 599373; doi: https://doi.org/10.1101/599373
Question 2.3. The result section for Immunohistochemistry data is missing. Have not mentioned about this.
Answer 2.3. We thank the Reviewer for pointing this out. The results section describing immunohistochemistry data is now included in the manuscript.
Question 2.4. Regarding the normalization of GluN1 IHC, is florescent value of anteroventral thalamic nucleus of both groups same? If that’s different then whole results will be affected. Also, are those experiments done all at once? Otherwise background florescence is different. Also, since it just has 2 subjects in alcoholic group, wont be reliable. It is not clear why authors could not have n=3 for this group. This data really not adding much besides showing an IHC of GluN1 which is also not great (background to signal noise is high). Representative western image is also not convincing if you look at grin1 blot.
Answer 2.4. We did our best to minimize the difference in background staining by doing the immunohistochemical stainings and analysis at the same time. The background staining is dependent on several parameters and mainly affected by post-mortem time before fixation. Autolytic processes start relatively fast in nervous tissue and could as well contribute to decrease in signal to noise ratio as epitopes are being degraded by catalytic processes. In order to gain any kind of quantitative information we measured fluorescence intensity from thalamic nuclei and substantia nigra pars reticulata and considered anteroventral thalamic nucleus to be “1.0”. Obtained results are shown in Figure 1. It is indeed a pitfall as changes in the fluorescence intensity of this nucleus could affect the results. Another good observation is considering the group size. As we are at the beginning of collecting human brain samples for immunohistochemical staining we did not gather enough human thalamic samples and our intention was not to draw any conclusions from these results. Therefore we made a bold assumption.
We have now included supplementary material showing the binding specificity of secondary antibodies. Signals obtained from these samples display low background fluorescence which is minimal. Localization of GRIN1 protein which is detected using validated antibody raised against human protein localizes as punta in nervous tissue, probably highlighting synapses. NeuN known marker for neuronal nuclei indeed localizes to nuclei and this is observable in our images.
Question 2.5. In Table 1, it would be helpful if authors include fold change of the gene in this study they are associated with GWAS studies.
Answer 2.5. We agree that fold change information makes Table 1 much more informative. The fold change information has now been added to the Table 1
Question 2.6. In Figure 3, it will be helpful for readers if they write down antibody with the color code on the top of image. Also, it is better to have a scale bar. Also figure legend is not clear: is it from control or alcholoic brain? It will be better if authors could put images from both subjects side by side for visual comparison.
Answer 2.6. It is true, the figure (Now Figure 1) would be easier to read with scale bar and colorcode - following modifications were included. The figure legend is supplemented with requested information. We decided to include asked information in the figure suplement (Supplementary figure S1).
Question 2.7. Also, I think it is helpful to have heat map image of highly altered DEGs on main manuscript or have a table of some of the highly altered DEGs so readers can look at it easily.
Answer 2.7. The heat map image of highly altered DEGs has now been included in the main manuscript (Figure 2) and the Table 1 that includes important DEGs supported by both earlier GWA studies and microarray data, is now completed with fold change information. We are thankful to the Reviewer for the comments and suggestions that significantly improved the quality of our manuscript.
Reviewer 3 Report
The authors examined postmortem differences in mediodorsal thalamic areas between individuals with alcohol-induced end-organ damage and controls. Findings suggested associations between history of alcoholism and upregulation of genes associated with glutamate and GABA, among others and some related to downregulation. A number of differentially expressed genes were those that have been identified in recent large-sample GWAS of alcohol use disorder or alcohol consumption phenotypes, linking these findings to specific genetic variants associated with drinking.
The manuscript is well written, and the link between differentially expressed genes and those found in recent GWAS strengthens the authors claims that alcohol-related molecular differences in the mediodorsal thalamus are associated with alcohol. The figures and tables present the complicated results in a clear manner.
I have some comments below for the authors to consider.
1. Can the authors explain how they can infer causation from these analyses? Specifically, languages is used throughout the manuscript suggesting that the observed differences reflect “obust molecular changes in the mediodorsal thalamus caused by alcohol.” It may be that I am not familiar enough with the techniques used in this manuscript, but it seems to me that observing differences between two groups, even in post-mortem analyses, are simply phenotypic differences that could be due to confounding (genetic pleiotropy between brain structure and disposition to alcohol use) or environmental exposure to an agent (ethanol). Genetically-informative study designs, like twin/family analyses or polygenic risk scores, can help disentangle etiology of an observed difference, but it’s not clear to me how finding differences between alcoholics and controls gives us evidence that it’s the alcohol (or a related environmental factor) that caused the differences.
2. The control group “did not have any signs of alcohol abuse, including alcohol-related organ damage and history alcohol use disorder (AUD) in medical records.” There is no mention of reviewing medical records of the alcoholic group to verify a diagnosis of AUD. Was this done, or is it feasible to do so? End-organ damage is a very face valid indicator of alcohol abuse, so a diagnosis of AUD perhaps is not necessary for the alcoholic group, but consistency between the groups would be nice.
3. In section 2.5, the authors state that they used as reference genes showing “associations with alcoholism“ in recent GWAS. However, some of the cited GWAS studies did not investigate alcoholism or a diagnosis of AUD, but rather drinking behaviors or alcohol consumption (e.g., Liu et al, 2019). It would be more accurate to state that the GWAS reference genes are from alcoholism and alcohol use phenotypes.
4. It would be interesting to see the results of the comparison of gene expression findings with recent GWAS studies analyses separated by genes related to AUD vs. alcohol consumption. Recent evidence suggests only partial genetic overlap between alcohol use disorder (AUD) and alcohol use (e.g., the Kranzler et al, 2019 cited by the authors), and so analyzing the set of GWAS-identified genes separately seems more informative to me than aggregating AUD and drinking-related genes.
5. As the authors acknowledge, the study sample size is small but they do find significant effects that survive FDR adjustment. However, they state several times that the study is “sufficiently powerful” to detect effects, and yet I see no power calculations or effect sizes. While the p-values are informative for the level of significance, presenting some metric of effect size (e.g., standardized beta, Cohen’s d between groups, etc.) seems like an important piece of information that is currently missing. Given the findings, I would assume the effect sizes must be fairly large to be this significant, and the small sample raises some concern regarding the power to detect potentially large effects. I understand that post-mortem data is difficult to collect relative to data in living individuals, so the small sample size is understandable, but more information about effect sizes and power calculations should be provided.
6. Figure 3 is the first figure referenced in the paper (on page 3), so why is it not Figure 1?
Author Response
Response to Reviewer 3 Comments:
Question 3.1. Can the authors explain how they can infer causation from these analyses? Specifically, languages is used throughout the manuscript suggesting that the observed differences reflect “robust molecular changes in the mediodorsal thalamus caused by alcohol.” It may be that I am not familiar enough with the techniques used in this manuscript, but it seems to me that observing differences between two groups, even in post-mortem analyses, are simply phenotypic differences that could be due to confounding (genetic pleiotropy between brain structure and disposition to alcohol use) or environmental exposure to an agent (ethanol). Genetically-informative study designs, like twin/family analyses or polygenic risk scores, can help disentangle etiology of an observed difference, but it’s not clear to me how finding differences between alcoholics and controls gives us evidence that it’s the alcohol (or a related environmental factor) that caused the differences.
Answer 3.1. It is true that case-control study designs are common in the field of public health but associations measured in these studies may or may not represent causal relationships. Conclusions drawn from case–control studies should be verified by replication in other designs (Melamed & Robinson, 2018). That is also the reason why we performed analysis of the expressional alterations in the context of recent GWAS studies. Recent GWA studies are powerful but GWAS data alone is also not providing much information about the molecular mechanisms. Therefore, one purpose of the current study was to point out molecular targets that are supported by both microarray and GWAS data deserve further focused research in the future studies of the genetics of alcohol abuse. This purpose has now been more clearly clarified in the “Conclusion” section (page 15 paragraph 2)
Melamed, A., and Robinson, J.N. (2019) Case-control studies can be useful but have many limitations: Study design: case-control studies. BJOG. 126(1):23. doi: 10.1111/1471-0528.15200.
Question 3.2. The control group “did not have any signs of alcohol abuse, including alcohol-related organ damage and history alcohol use disorder (AUD) in medical records.” There is no mention of reviewing medical records of the alcoholic group to verify a diagnosis of AUD. Was this done, or is it feasible to do so? End-organ damage is a very face valid indicator of alcohol abuse, so a diagnosis of AUD perhaps is not necessary for the alcoholic group, but consistency between the groups would be nice.
Answer 3.2. Thank you for pointing this out and giving us the opportunity to clarify our selection and grouping process. Available medical records were identical for both our groups. Previous medical records were available for both groups using our Electronic Health Record (e-Health Record). Lifetime alcohol consumption from next-of-kin interviews was not available.
Subjects in our alcohol group were selected based on prior diagnosis of alcohol related disorder together with clinical diagnosis of alcohol related end-organ damage supported by post-mortem histology. Subjects in our non-alcoholic group had no prior diagnosis of alcohol related disorder, end-organ damage and presented no signs of alcohol related damage collected from tissue samples taken during the autopsy.
We have accordingly modified (from end of page 2 to the first paragraph in page 3) to emphasize this point:
Chronic alcohol abuse of the deceased (n=15) was based on prior diagnosis of alcohol related disorder together with clinical diagnosis of alcohol related end-organ damage supported by post-mortem histology, including alcoholic liver disease, alcoholic pancreatitis, and alcoholic cardiomyopathy. Non-alcoholic study group (n=12) included subjects which had no prior diagnosis of alcohol related disorder code, end-organ damage, and signs of alcohol related damage collected from tissue samples taken during the autopsy.
Question 3.3. In section 2.5, the authors state that they used as reference genes showing “associations with alcoholism“ in recent GWAS. However, some of the cited GWAS studies did not investigate alcoholism or a diagnosis of AUD, but rather drinking behaviors or alcohol consumption (e.g., Liu et al, 2019). It would be more accurate to state that the GWAS reference genes are from alcoholism and alcohol use phenotypes.
Answer 3.3. We thank the reviewer to point out an important issue to clarify the exact GWAS phenotypes (mainly AUD or alcohol consumption measurement), which have now been shown in the Supplementary table S3.
Question 3.4. It would be interesting to see the results of the comparison of gene expression findings with recent GWAS studies analyses separated by genes related to AUD vs. alcohol consumption. Recent evidence suggests only partial genetic overlap between alcohol use disorder (AUD) and alcohol use (e.g., the Kranzler et al, 2019 cited by the authors), and so analyzing the set of GWAS-identified genes separately seems more informative to me than aggregating AUD and drinking-related genes.
Answer 3.4. Specific GWAS phenotypes along with corresponding p-values have now been added to the Supplementary table S3. Interestingly, as it now appears, most of the genes supported by both gene array and GWAS data were associated with both alcohol use disorder and alcohol consumption phenotypes, such as DRD2, FTO, TCF4, CADM2, CRHR1, synaptotagmin 14, ADH1B and GCKR. This information is now included to the end of “Results” section 3.3. (page)
Question 3.5. As the authors acknowledge, the study sample size is small but they do find significant effects that survive FDR adjustment. However, they state several times that the study is “sufficiently powerful” to detect effects, and yet I see no power calculations or effect sizes. While the p-values are informative for the level of significance, presenting some metric of effect size (e.g., standardized beta, Cohen’s d between groups, etc.) seems like an important piece of information that is currently missing. Given the findings, I would assume the effect sizes must be fairly large to be this significant, and the small sample raises some concern regarding the power to detect potentially large effects. I understand that post-mortem data is difficult to collect relative to data in living individuals, so the small sample size is understandable, but more information about effect sizes and power calculations should be provided.
Answer 3.5. We agree that it is overstatement to call current study “sufficiently powerful”. Due to the limited access to post-mortem tissues, we were not able to perform a pilot study evaluating the potential amount of variation in the data, which is needed for the power calculations. Also the minimal effect size, which would be clinically or scientifically meaningful is hard to evaluate at the current knowledge stage. Thus, with a small sample available for us we expected to see an occurrence of medium or large effects in the thalamic area of alcoholic subjects versus controls. The section pointing out this methodological limitation has now been added to the “Materials and methods” section (page 4, paragraph 2), the statements that the study is “sufficiently powerful” have now been removed from the “Abstract” and “Conclusions”
Question 3.6. Figure 3 is the first figure referenced in the paper (on page 3), so why is it not Figure 1?
Answer 3.6. Indeed, earlier Figure 3 has been referred first in the manuscript, thus, the order of the figures has now been changed and corresponds to the order of reference.
Round 2
Reviewer 2 Report
Most of my concerns are addressed and the manuscript looks better now than first submission. I still prefer qRT-PCR for validation of gene expression in Chip assay. Other than that I am satisfied with author's response.
Author Response
We understand the significance of the suggestion to use qRT-PCR for validation of gene expression in Chip assay. The following sentence has now been added to the “Results” section (page 10, paragraph 1): It is important to note that transcript levels and protein levels are not always highly correlated therefore qRT-PCR would have been preferred method for chip assay validation.